# Deep Sturm–Liouville: From Sample-Based to 1D Regularization with Learnable Orthogonal Basis Functions

**David Vigouroux** [1 2]  **Joseba Dalmau** [1 2 3]  **Louis Bethune** [2 4]  **Victor Boutin** [5 2]

## Abstract

Although Artificial Neural Networks (ANNs) have achieved remarkable success across various tasks, they still suffer from limited generalization. We hypothesize that this limitation arises from the traditional sample-based (0–dimensionnal) regularization used in ANNs. To overcome this, we introduce *Deep Sturm–Liouville* (DSL), a novel function approximator that enables continuous 1D regularization along field lines in the input space by integrating the Sturm–Liouville Theorem (SLT) into the deep learning framework. DSL defines field lines traversing the input space, along which a Sturm–Liouville problem is solved to generate orthogonal basis functions, enforcing implicit regularization thanks to the desirable properties of SLT. These basis functions are linearly combined to construct the DSL approximator. Both the vector field and basis functions are parameterized by neural networks and learned jointly. We demonstrate that the DSL formulation naturally arises when solving a Rank-1 Parabolic Eigenvalue Problem. DSL is trained efficiently using stochastic gradient descent via implicit differentiation. DSL achieves competitive performance and demonstrate improved sample efficiency on diverse multivariate datasets including high-dimensional image datasets such as MNIST and CIFAR-10.

## 1. Introduction

Neural networks have become the go-to approach in various applications, demonstrating their versatility in a wide range of tasks from image recognition to natural language

[1]IRT Saint Exupery [2]ANITI (Artificial and Natural Intelligence Toulouse Institute) [3]IMT, INSA Toulouse [4]Now at Apple [5]CerCo (CNRS, Université de Toulouse). Correspondence to: David Vigouroux <david.vigouroux@irt-saintexupery.com>.

*Proceedings of the 42$^{nd}$ International Conference on Machine Learning*, Vancouver, Canada. PMLR 267, 2025. Copyright 2025 by the author(s).

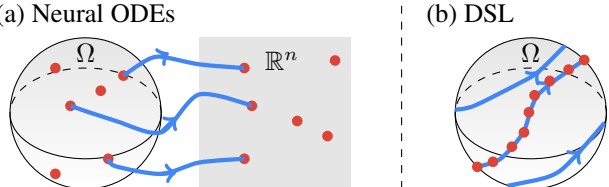

(a) Neural ODEs     (b) DSL

*Figure 1.* **(a)** In Neural ODEs, the vector field acts as a continuous-depth neural network. Regularization applied along a field line represents a *0D* regularization (i.e. sample specific). **(b)** For DSL, the vector field's field lines span the entire input space. Regularizing along these lines applies to all points they pass through, making it *1D* regularization.

processing. These practical results are also supported by theoretical works on the expressivity of neural networks. It has long been known that any function can be approximated by neural networks (Hornik et al., 1989; Cybenko, 1989) and recent works demonstrate exponential approximation accuracy (Elbrächter et al., 2021).

Despite its remarkable achievements, deep learning still suffers from significant generalization limitations. For example, neural networks are vulnerable to adversarial attacks, where small changes to the input can drastically alter predictions (Moosavi-Dezfooli et al., 2015). Another critical issue is domain shift, which occurs when a model trained on a specific data distribution fails to generalize to new but related data —such as images captured under different lighting conditions, viewpoints, or backgrounds— despite their semantic similarity (Tzeng et al., 2014; Ganin et al., 2016; Ben-David et al., 2010). In addition, Neural Network often struggle to generalize to testing samples when trained with a limited number of samples (Zhang et al., 2021; Arpit et al., 2017; McClellan et al., 2024). These generalization issues have spurred entire fields of active research (Rodriguez et al., 2023; Linsley et al., 2023; Szegedy et al., 2013), yet their root causes remain poorly understood. Do they stem from the optimization process, the learning paradigm, the regularization techniques, the network architectures, or other underlying factors?

In this work, we argue that the sample-based approach commonly used in deep learning to compute or regularize the

loss function may be a fundamental limitation. Specifically, conventional sampling-based regularization—referred to here as $0D$ regularization—evaluates the function at discrete points within the domain space, inherently capturing only a partial view of that space. Our core contribution is to move beyond this traditional approach by introducing $1D$ regularization, a continuous regularization method applied along trajectories that span the entire domain space. Unlike $0D$ regularization, which focuses on isolated points, $1D$ regularization captures richer structural information by integrating along continuous paths. Our work complements other regularization methods such as weight decay or per-sample regularization (e.g. entropy regularization (Grandvalet & Bengio, 2004) or KL regularization (Higgins et al., 2016)). However, unlike traditional approaches, DSL emphasizes function smoothness by enforcing regularization along continuous trajectories in the input space.

To enable 1D regularization, we introduce a novel class of function approximators, termed Deep Sturm–Liouville (DSL), which possess unique properties. DSL is built upon an ODE defining a vector field (Feldman & Yeager, 2018), parameterized by a neural network, that guides trajectories across the input space. Along each trajectory —specifically, each field line of the vector field— the function is approximated using an orthogonal basis, obtained by solving the Sturm–Liouville (SL) Eigenvalue Problem. The first few basis functions of the SL problem exhibit desirable *regularity* properties, which naturally impose implicit regularization along the entire field line. Moreover, the DSL framework is flexible enough to integrate explicit constraints along the field line. Unlike other ODE-based frameworks, such as neuralODE (Chen et al., 2018b), our approach leverages field lines that are specifically designed to traverse the entire input space (see Fig. 1). This design enables DSL to effectively apply $1D$ regularization, ensuring a more structured constraint across the input domain.

We demonstrate that this new function approximator is intrinsically connected to a more general mathematical framework, the rank-1 Parabolic Eigenvalue Problem. This connection is particularly significant as it shows that the DSL formulation emerges naturally as a solution to the rank-1 Parabolic Eigenvalue Problem.

First, we introduce the Sturm-Liouville theorem (SLT) in its original 1D form. Then, we expose the Elliptic Eigenvalue Problem which extends SLT to the multidimensional case. This Elliptic Eigenvalue Problem being hard to solve due to its significant calculation time, we also present a related problem called Parabolic Eigenvalue Problem. In section 4, we introduce the Deep Sturm–Liouville method, which exploits a link between the Rank-1 Parabolic Eigenvalue Problem and the Sturm–Liouville problem to obtain a tractable solution for the high-dimensional case. Then,

we describe how to introduce 1D regularization within this framework. We finish with an experimental evaluation of the DSL method where we demonstrate experimentally the expressivity of the method.

Our main contributions are:

- Introducing a new function approximator, called *Deep Sturm–Liouville* (DSL), built from a task-dependent orthogonal function basis in an open domain. We demonstrate experimentally the expressivity of DSL.
- Formulating 1D regularization within the DSL framework to control the solution smoothness through implicit and explicit spectral conditioning.
- Establishing a link between Deep Sturm–Liouville and a Rank-1 Parabolic Eigenvalue Problem.

## 2. Related work

Neural Ordinary Differential Equations (Neural ODEs), introduced by (Chen et al., 2018b), parameterize the continuous dynamics of hidden units using an ODE specified by a neural network. Unlike traditional discrete-layer architectures, Neural ODEs provide a continuous representation and have been widely applied in fields such as normalizing flows (Tabak & Vanden-Eijnden, 2010; Rezende & Mohamed, 2015). In contrast, DSL is not a continuous-depth neural network; instead, it introduces a novel function approximator based on orthogonal bases. The primary difference between DSL and continuous-depth neural networks lies in their vector fields (see figure 1). In continuous-depth networks, the vector field maps the input space to the state space ($\Omega \to \mathbb{R}^n$). Therefore in Neural ODE the regularization is applied on a per-sample basis ($0D$). Conversely, DSL's vector field operates directly within the input space ($\Omega \to \Omega$), ensuring that field lines traverse the entire domain. This enables a $1D$ regularization approach that optimizes the loss along entire trajectories rather than just on individual samples, improving generalization through implicit regularization. DSL achieves this by controlling the number of sign changes in the basis function.

Continuous-depth neural network have also gained significant attention in the context of generative modeling, particularly in applications such as normalizing flows introduced by Tabak & Vanden-Eijnden (2010) and popularised by Rezende & Mohamed (2015). These models leverage continuous dynamics to transform probability distributions, thanks to the Liouville Identity, enabling flexible and efficient mapping between complex data distributions and simpler base distributions. Normalizing flows provide a powerful framework for density estimation, generative sampling, and latent space exploration. Integrating DSL into such a framework requires overcoming significant challenges, as the change-of-variable formula used in our method is funda-

mentally different from those typically employed in Neural Odes. This difference in the change of variable prevents to compute density estimation with our work. DSL should be regarded solely as a function approximator.

While Neural ODEs operate with a fixed integration time, (Massaroli et al., 2020) introduces Adaptive-Depth Neural ODEs, where the integration time of the ODE is learned through a neural network. In Deep Sturm–Liouville, the integration time also changes for every sample. The integration is obtained by a stopping criteria when the trajectory cross the boundary of the domain. The stopping condition of the trajectory is governed by a spatial event condition instead of being determined temporally by a neural network.

The simplest approach to learning an orthogonal basis is through the Gram-Schmidt process. However, unlike Gram-Schmidt, which defines orthogonality locally in the neighborhood of a sample, DSL enforces orthogonality along each field line. By extension, DSL maintains orthogonality across the entire domain $\Omega$, as stated in Theorem 5.1.

Some works solve partial differential equations with neural networks, e.g. for magnetic field estimation (Khan et al., 2019), fluid simulations (Kim et al., 2019), eigenvalue functions problems (Kovacs et al., 2022) or PDEs similar to the Elliptic Eigenvalues Problems (Marwah et al., 2023). Inspired by complex natural phenomena which can be simulated by PDEs, our work takes the opposite point of view of these works by using Rank-1 Parabolic Eigenvalue Problems to propose a new function approximator.

# 3. Notation and Eigenvalues Problems

We define the data open domain $\Omega \subset \mathbb{R}^n$ with targets $Y \in \mathbb{R}^k$. We assume that $P_{XY} = \mathcal{P}(\Omega \times \mathbb{R}^k)$ is the joint distribution of data points and targets, and we consider a dataset $\mathcal{D} \sim P_{XY}^{\otimes n}$. We define the predictor $F$ as a pair $(\theta, L)$ composed of a vector of orthogonal basis functions $\boldsymbol{u^\theta} : \Omega \to \mathbb{R}^d$ (d is a hyper-parameter), parametrized by $\theta$ with a weight function $w^\theta : \Omega \to \mathbb{R}_*^+$, and a linear map $L : \mathbb{R}^d \to \mathbb{R}^k$:

$$F(x, \theta, L) \stackrel{\text{def}}{=} L(\boldsymbol{u^\theta}(\boldsymbol{x})),$$
$$\text{s.t} \int_\Omega w^\theta(x) \boldsymbol{u}_i^{\boldsymbol{\theta}}(x) \boldsymbol{u}_j^{\boldsymbol{\theta}}(x) dx = 0 \quad \forall i \neq j. \tag{1}$$

Our goal is to minimize the empirical risk associated to a loss $\mathcal{L} : \mathbb{R}^k \times \mathbb{R}^k \to \mathbb{R}$, defined as

$$\min_{\theta, L} \frac{1}{n} \sum_{i=1}^n \mathcal{L}\big(y_i, F(x_i, \theta, L)\big) \tag{2}$$

by simultaneously learning the linear operator $L$ and the orthogonal function basis $\boldsymbol{u^\theta}(\boldsymbol{x})$, in a *data-dependent fashion*. To avoid the curse of dimensionality incurred by fixed basis functions such as Fourier, polynomial or wavelet, the

aim of this work is to create a flexible framework where the orthogonal basis functions are not fixed but they are learnt to adapt to a particular machine learning task.

In this work, to obtain the orthogonal functions $u_i(x)$, we will study a special case of eigenvalue problem: The Sturm–Liouville Problem (SLP). An eigenvalue problem involves finding scalars ($\lambda$, eigenvalues) and corresponding vectors ($v$, eigenvectors) such that for a given linear operator or matrix $A$, the equation $Av = \lambda v$ holds, where $v \neq 0$. In the case of SLP, the eigenvectors are 1D function vectors that form a complete basis. These functions exhibit desirable regularity properties and are themselves parameterized by functions, which provide significant flexibility compared to fixed bases. An extension of this theorem exists for the general case in N-D: the Elliptic Eigenvalue Problem. However, this problem becomes intractable in high dimensions. To address this, we will study a more tractable solution: the 1-rank Parabolic Problem, which is a *degenerate* case of the Elliptic Eigenvalue Problem.

## 3.1. Sturm–Liouville theorem

The Sturm–Liouville theorem (Sturm & Liouville, 1837) has a significant importance on the theory of eigenvalue problems for 1D ordinary differential equations (ODE). For instance, Sturm–Liouville theory (SLT) is employed in quantum mechanics to analyze the solutions of the Schrödinger equations (Bender & Orszag, 1978), in heat conduction problems (Lützen, 1984) or to compute vibrational modes (Wang, 1996). Sturm–Liouville eigenvalue problems offer a systematic approach to discerning the characteristic frequencies and spatial patterns. This relationship between SLT and physics problems motivates us to explore the potential application of this theorem in machine learning. A wide range of 1D complete orthonormal function bases can be reinterpreted within this theory; common bases such as Fourier, Bessel or Chebyshev polynomials can be seen as particular cases of this setting.

The Sturm–Liouville theorem is formulated as an eigenvalue and eigenfunction problem satisfying the boundary conditions of an ODE:

**Theorem 3.1** (Sturm–Liouville Theorem). *For any given functions, $p, w : [a, b] \to \mathbb{R}_0^+$ and $q : [a, b] \to \mathbb{R}$ of classes $\mathcal{C}^1$, $\mathcal{C}^0$ and $\mathcal{C}^0$ respectively, and real numbers $\alpha_1, \alpha_2, \beta_1, \beta_2$, there exist a unique sequence $\{\lambda_i\}_{i \geq 1}$ (of eigenvalues) and associated eigenfunctions $y_i : [a, b] \to \mathbb{R}$ solving the ODE below, with the given boundary conditions:*

$$-\frac{d}{dt}\left[p(t)\frac{dy_i(t)}{dt}\right] + q(t)y_i(t) = \lambda_i w(t)y_i(t),$$
$$\alpha_1 y_i(a) + \alpha_2 \frac{dy_i}{dt}(a) = 0 \qquad \alpha_1, \alpha_2 \text{ not both 0}, \tag{3}$$
$$\beta_1 y_i(b) + \beta_2 \frac{dy_i}{dt}(b) = 0 \qquad \beta_1, \beta_2 \text{ not both 0}.$$

The sequence of eigenfunctions $\{y_i(t)\}$ forms an orthonormal basis in the Hilbert space $L^2([a,b])$ with the inner product weighted by $w$:

$$\int_a^b w(t)y_i(t)y_j(t)dt = \delta_{ij}. \qquad (4)$$

where $\delta$ is the kronecker delta.

The eigenvalues $\lambda_1, \lambda_2, ...$ are real and ordered so that $\lambda_1 < \lambda_2 < ...\lambda_n < ... \to \infty$. According to Egorov & Kondratiev (1996) Chapter 5-Theorem 19, the $n^{th}$ basis function has exactly $n-1$ zeros in the interval $]a,b[$. By linearly combining the first $n$ basis functions to construct a function approximator and tuning $n$, we can control the level of regularity of the function. Herein, we assume Dirichlet's boundary conditions: $y_i(a) = 0$ and $y_i(b) = 0$.

For one dimensional data, we can parameterize the functions $p$, $q$ and $w$ with neural networks. By solving the associated Sturm–Liouville Problem, we obtain the eigenfunctions $y_i(t)$ which form an orthogonal basis. A linear combination of the $y_i(t)$ can be used to predict the value at any $x \in [a,b]$. The weights of $p, q, w$ can be learnt to optimize (2). The main idea behind our work is to extend this procedure to the multidimensional case.

### 3.2. Elliptic Eigenvalue Problem

The Strum-Liouville theorem has its extension in dimension greater than one, more precisely on an open set $\Omega$, thanks to the following Elliptic Eigenvalue Problem (EEP) (Larsson, 2003; Muthukumar, 2014):

**Theorem 3.2.** *For any continuous functions $A : \Omega \to \mathbb{R}^n \times \mathbb{R}^n$, symmetric, positive-definite, $q : \Omega \to \mathbb{R}$ and $w : \Omega \to \mathbb{R}_+^*$ of classes $\mathcal{C}^1$, $\mathcal{C}^0$ and $\mathcal{C}^0$ respectively, there exist a unique sequence of eigenvalues $\lambda_i$ and associated eigenfunctions $u_i$ satisfying:*

$$\nabla \cdot (A(x) \cdot \nabla u_i(x)) + q(x)u_i(x) = -\lambda_i w(x)u_i(x).$$
$$\text{with } u_i(x) = 0 \qquad \forall x \in \partial\Omega. \quad (5)$$
$$\int_\Omega w(x)u_i(x)u_j(x)dx = \delta_{ij}.$$

This theorem could be useful to learn a basis of functions suited to a particular machine learning task in high dimension by optimizing Eq. 2 through the optimization of the functions $A$, $q$ and $w$, typically surrogated by neural networks.

Solving these equations directly is quite challenging. First, even if recent works study the solutions of partial differential equations in high dimension (Wu et al., 2023), efficiently solving this kind of partial differential equations (PDE) is very costly. Secondly, solving the eigenvalue problem is very difficult even if numeric methods exists (Larsson,

2003). Thirdly, without assumptions on the form of the matrix $A$, whose size is the square of the size of input space, the matrix $A$ can be too large for high dimension data.

To make this computational problem more tractable, we explore a related problem where the rank of the matrix $A$ is equal to 1 to simplify the problem structure. We refer to this new problem as the Rank-1 Parabolic Eigenvalue Problem, defined as:

**Definition 3.3.** The Eigenvalue Problem (see Eq. 5) is called Rank-1 Parabolic Eigenvalue Problem when the matrix A is positive semi-definite and its rank is equal to 1.

In such case, the existence of eigenvalues is not guaranteed. However, its rank-1 structure will allow us to solve the Parabolic Eigenvalue Problem along a field line by using the 1D Sturm–Liouville theorem (see Theorem 5.3). This will allow us to combine the Sturm–Liouville theorem and deep neural networks, thus giving rise to Deep Sturm–Liouville, a means to compute orthogonal bases in high dimensions without the need to solve a high dimensional PDE.

## 4. Deep Sturm–Liouville Method

Deep Sturm–Liouville (DSL) is constructed using a vector field that traverses an open domain $\Omega \subset \mathbb{R}^n$. Along each field line, a Sturm–Liouville problem is solved to generate orthogonal basis functions. When linearly combined, these basis functions define a new function. Machine learning problems formulated as the optimization problem in Eq. 2 can be solved using gradient descent.

### 4.1. Deep Sturm–Liouville

First, we define the field line $\gamma^x(t) = z(t)$ associated with the sample $x$, which satisfies the following equation parameterized by the function $a : \Omega \to \mathbb{R}^n$:

$$\frac{dz}{dt} = a(z), \quad z(0) = x. \qquad (6)$$

Note that Eq. 6 is similar to the one used in Neural Ordinary Differential Equations (Chen et al., 2018a). However, in this work, this equation serves a different purpose: to project the sample distribution onto the boundaries of the domain $\Omega$. While Neural ODEs have fixed final time, Deep Sturm–Liouville assigns a unique final time for each sample $x$. For any $x \in \Omega$, we enforce that the field line $\gamma^x(t)$ passing through $x$ intersects the boundary $\partial\Omega$ in two unique points to ensure complete coverage of the domain $\Omega$. That is, There exist two unique times $t_-^x < 0$ and $t_+^x > 0$ such that $\gamma^x(t_-^x)$ and $\gamma^x(t_+^x) \in \partial\Omega$.

To obtain the uniqueness and the existence of $t_-^x$ and $t_+^x$, we assume that Eq. 6 has an unique solution – which requires $a(x)$ to be Lipschitz continuous –, that there are no limit cycles and that $a(x)$ is nowhere tangent to $\partial\Omega$. The existence

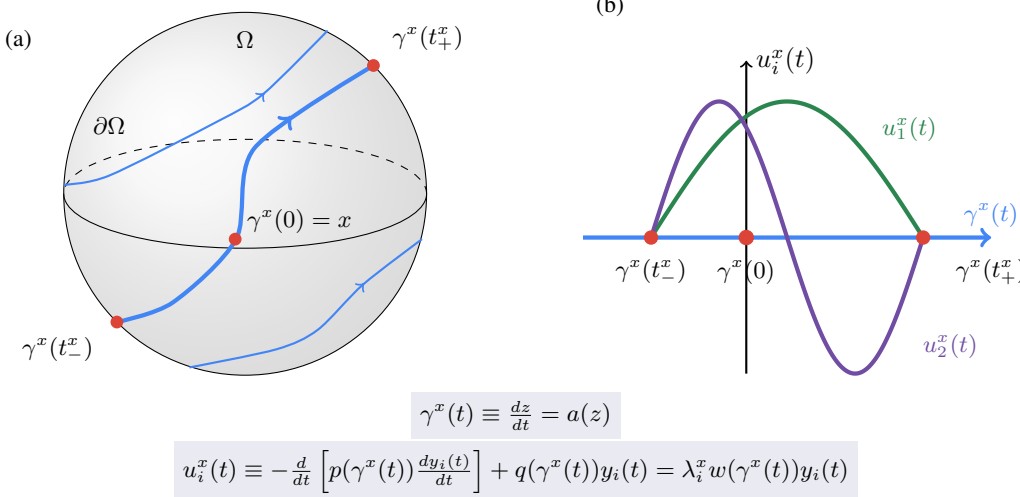

$$\gamma^x(t) \equiv \frac{dz}{dt} = a(z)$$

$$u_i^x(t) \equiv -\frac{d}{dt}\left[p(\gamma^x(t))\frac{dy_i(t)}{dt}\right] + q(\gamma^x(t))y_i(t) = \lambda_i^x w(\gamma^x(t))y_i(t)$$

*Figure 2.* **Deep Sturm–Liouville.** – **(a)** For a given point $x$, the field line $\gamma^x(t)$ is defined by equation (6), it is such that $\gamma^x(0) = x$ and reaches the two points at the boundary of $\Omega$ at time $t_-^x$ and $t_+^x$. **(b)** On the field line $\gamma^x(t)$, the Sturm–Liouville Problem 7 is solved with the parameter functions $p(\gamma^x(t))$, $q(\gamma^x(t))$ and $w(\gamma^x(t))$ to obtain orthogonal function basis $u_i^x$ that, combined linearly, form the DSL function approximator along the field line. The prediction at x is obtained by taking the value of this function at $t = 0$.

of limit cycles is a complex problem for which no general solution is known for $n > 2$; for n=2 the Bendixson–Dulac theorem describes sufficient conditions to have no limit cycles (Burton & Burton, 1983). However, special cases exists where the absence of limit cycles has been demonstrated. For example (Johnston, 2015):

- $a$ is a strictly positive continuous function and $\Omega$ is convex,
- $a$ is a gradient of a function with no singular point and the gradient is not vanishing anywhere in the domain[1].

The field line $\gamma^x(t)$, defined by Eq. 6, is restricted to the temporal interval $[t_-^x, t_+^x]$. Along this segment of the field line, a Sturm–Liouville eigenvalue problem is solved to construct a 1D orthogonal basis specific to this part of the domain $\Omega$. The Sturm–Liouville Problem is parametrized by the learnable functions $p : \Omega \to \mathbb{R}_*^+$, $q : \Omega \to \mathbb{R}$, $w : \Omega \to \mathbb{R}_*^+$ and $v : \partial\Omega^- \to \mathbb{R}^d$. These functions are optimized during the training phase to generate adaptive basis functions tailored to the problem. Formally, the 1D orthogonal basis functions $u_i^x(t)$ and their corresponding eigenvalues $\lambda_i^x$ are obtained by solving the following eigenvalue problem:

$$-\frac{d}{dt}\left[p(\gamma^x(t))\frac{du_i(t)}{dt}\right] + q(\gamma^x(t))u_i(t) = \lambda_i^x w(\gamma^x(t))u_i(t),$$

$$u_i(t_-^x) = 0, \qquad u_i(t_+^x) = 0, \qquad \frac{du_i(t_-^x)}{dt} = v_i(\gamma^x(t_-^x)). \tag{7}$$

*Remark* 4.1. The ode and the Dirichlet's conditions of Eq. 7

---
[1]This equation has some similarities with energy-based models (Hinton, 2002).

are defined up to a multiplying coefficient. The equation on the $u_i^x$ derivative at $t_-$ ensures the uniqueness of the solution. $v(x)$ is learnable; however, in most experiments, a fixed value of $v(x) = 1$ proves to be sufficient.

We define the function $u_i : \Omega \to \mathbb{R}$ by setting $u_i(x) = u_i^x(0)$. The $u_i(x)$ are well defined and form an orthogonal basis along the field line $\gamma^x$. Indeed, for all $x_1 \in \Omega$, if $x_2 = \gamma^{x_1}(s)$, we can show that $u_i^{x_1}(t) = u_i^{x_2}(t - s)$. As a consequence:

$$u_i(\gamma^{x_1}(s)) = u_i(x_2) = u^{x_2}(0) = u^{x_1}(s), \forall s \in [t_-^{x_1}, t_+^{x_1}] \tag{8}$$

This implies that the integral Eq. 4 can be rewritten as a function of the field-line:

$$\int_{t_-^x}^{t_+^x} w(\gamma^x(t))u_i(\gamma^x(t))u_j(\gamma^x(t))dt = 0.$$

*Remark* 4.2. As demonstrated in Theorem 5.1, the collection of 1D bases combines to form a global orthogonal basis over the entire domain $\Omega$.

In the Sturm–Liouville Theorem, the functions $p$, $q$ and $w$ depend only on the variable $t$. In Deep Strum-Liouville, the key idea is that $p$, $q$ and $w$ depend on the field line $\gamma^x(t)$. The purpose of this dependence is to couple Eq. 6 and Eq. 7 through the variable $t$. For two samples $x^1$ and $x^2$, which belong to two different field lines $\gamma^{x^1}$ and $\gamma^{x^2}$, two different *local* orthogonal 1D basis functions are estimated. Consequently, the function approximator on the whole domain $\Omega$ is composed of 1D basis functions which are locally orthogonal.

Once the eigenvalues are obtained, the prediction at a given $x$ is computed by solving the ordinary differential equations to derive $u_i^x(0)$ from $t_-^x$ (see Eq. 7). The function approximator is then defined through a linear map $L : \mathbb{R}^d \to \mathbb{R}^k$, where $d$ represents the number of eigenvalues (a tunable parameter of our method), and $k$ denotes the dimensionality of the predictor's output $F$.

$$F^{\theta,L}(x) = L\left(\boldsymbol{u}^\theta(x)\right) \quad \text{with } \theta = [a, p, q, w, v].$$

---

**Algorithm 1** Deep Sturm–Liouville - Prediction
1: Compute $t_-^x$ and $t_+^x$ with Eq. 6
2: Find eigenvalues $\lambda_i^x$ along the field line $\gamma^x(t)$ in (7) using (A)
3: Resolve Eq. 7 from $t_-^x$ to compute $u_i(x)$
4: Compute the prediction at $x$: $F^{\theta,L}(x) = L\left(\boldsymbol{u}^\theta(x)\right)$

---

The optimization problem in Eq. 2 can be rewritten by minimizing the parametric functions of the Sturm–Liouville problem:

$$\min_{L,\theta} \mathcal{L}(Y, F^{\theta,L}(X)).$$

In our experiments, the functions $a(x)$, $p(x)$, $q(x)$, $w(x)$, $v(x)$ are typically parameterized by neural networks.

### 4.2. Regularizations

The central idea of Deep Sturm–Liouville is to introduce a $1D$ regularization along each field line traversing the input space $\Omega$, aiming to achieve better generalization compared to the sample-based regularization (i.e. $0D$) typically used in machine learning. This $1D$ regularization is both implicit and explicit:

**Implicit regularization:** This regularization is a natural consequence of the Sturm–Liouville Theorem. It is obtained by selecting the first few elements of the basis. The first elements of the DSL basis *oscillate* less than higher-ranked elements of the basis (similar to the Fourier basis functions). In the Sturm–Liouville framework, the *oscillation* is defined by the number of times where the basis functions change sign: the $n^{th}$ base function changes sign exactly $n-1$ times. By selecting the $d$ first elements of the basis function, DSL guarantees an implicit regularization along each field line.

**Explicit regularization – Spectral regularization:** To avoid strong variations on the derivatives of the basis along the field line $\gamma^x$, the absolute value of the eigenvalues of the Sturm–Liouville Theorem, computed for each field line $\gamma^x(t)$, are added in the loss as a regularization term:

$$\min_{L,\theta} \mathbb{E}\left(\mathcal{L}(Y, F^{\theta,L}(X)) + \frac{\alpha}{d}\sum_{i=0}^{d}|\lambda_i(X)|\right). \quad (9)$$

### 4.3. Algorithm details

**Eigenvalues** To compute the eigenvalues of the Sturm–Liouville problem (see Eq. 3), a shooting method (Stoer et al., 2002) is performed. The aim of the shooting method is to optimize the eigenvalue $\lambda$ such that the boundary condition $y(b) = 0$ is satisfied. In our work, we perform a binary search between the lower and upper bounds of the eigenvalues (Breuer & Gottlieb, 1971) on an equivalent problem obtained by the Prüfer Substitution (Prüfer, 1926; Lebovitz, 2019), see appendix A for more details.

**Gradients** The computation of the gradients of Deep Sturm–Liouville w.r.t the weights of the function $a$, $p$, $q$ and $w$ is not straightforward due to the estimation of eigenvalues the $\lambda_i^x$ and the times $t_-^x$ and $t_+^x$ associated to each prediction. In fact, the computation through the shooting process and the stop conditions are not differentiable. To overcome this, we use the implicit differentiation theorem (Krantz & Parks, 2012) thanks to a mapping function capturing the optimal conditions of the problem (see more details in appendix B).

## 5. Results

### 5.1. Theoritical Results

**Deep Sturm–Liouville is an orthogonal basis on $\Omega$:** Let us state the main theorem of Deep Sturm–Liouville:

**Theorem 5.1.** *The functions $u_i(x)$ form an orthogonal basis of functions on the open domain $\Omega$ with a weight function $w^*(x) : \Omega \to \mathbb{R}_*^+$:*

$$\int_\Omega w^*(x)u_i(x)u_j(x)dx = 0.$$

The intuition behind the proof of this theorem is simple. Along the field line $\gamma^x$ the basis functions $u_i(x)$ are orthogonal. By applying a Fubini-like (Nicolaescu, 2011) result to the integral over the whole domain $\Omega$, we rewrite the integral over $\Omega$ as a double integral: over the points in the boundary $\partial\Omega$ of the form $\gamma^x(t_-^x)$, and along the field line $\gamma^x$, thus obtaining the orthogonality over the whole domain $\Omega$ (see proof in appendix C).

**Deep Sturm–Liouville is a relaxation of the Rank-1 Parabolic Eigenvalue Problems:** Interestingly, Deep Sturm–Liouville originated from studying solutions of the Elliptic Eigenvalue Problem. What stands out is that DSL was discovered as a natural outcome of addressing this broader mathematical problem, rather than being a clever combination of existing techniques. Deep Sturm–Liouville can be seen as a relaxed version of the Rank-1 Parabolic Eigenvalues Problem when no *global* eigenvalues exists. We define a sub-class of Deep Sturm–Liouville problems.

**Definition 5.2.** Deep Sturm–Liouville Problem (see Eq. 7) is uniform if all eigenvalues are independent of $x$.

**Theorem 5.3.** *The Uniform Deep Sturm–Liouville Problem can be rewritten as a Dirichlet Rank-1 Parabolic Eigenvalue Problem when assuming $a(x)$ is the gradient of a function with $a(x)_i > 0$ and by defining $v_i : \mathbb{R} \times \mathbb{R}^{n-1} \to \mathbb{R}$.*

$$\nabla \cdot (a(x)a^t(x) \cdot \nabla u_i(x)) + q(x)u_i(x) = -\lambda_i w(x)u_i(x).$$

$$\Leftrightarrow \begin{cases} \frac{\partial}{\partial t}\left(p(x)\frac{\partial v_i(t,\boldsymbol{y})}{\partial t}\right) + \tilde{q}(x)v_i(t,\boldsymbol{y}) = -\lambda_i \tilde{w}(x)v_i(t,\boldsymbol{y}), \\ \frac{dx}{dt} = \tilde{a}(x). \end{cases}$$

The idea is to define a direction $a(x)$ in which the PDE could lead to a more tractable ODEs. This could be done by rewriting the positive semi-definite rank$-1$ matrix $A(x)$ as $A(x) = a(x)a^t(x)$ and by applying a change of variables where all gradients of this new system of variables are orthogonal to $a(x)$ (see proof in appendix D).

## 5.2. Experiments

To evaluate our work, Deep Sturm–Liouville has been trained on three multivariate datasets: Adult (Becker & Kohavi, 1996), Dry Bean (UCI Machine Learning Repository) and Bank Marketing (Frank, 2010), as well as the MNIST image dataset (LeCun & Cortes, 2010) and the Cifar10 dataset (Krizhevsky et al., 2009). Due to the fundamental differences in the change of variables between continuous neural networks and DSL, no experiments were conducted for density estimation.

Deep Sturm–Liouville is implemented on `jax` (Bradbury et al., 2018) and uses `diffrax` (Kidger, 2021) to solve the ODEs involved. The solver `dopri8` (Prince & Dormand, 1981) is used with a relative tolerance of $1\mathrm{e}{-6}$ and an absolute tolerance of $1\mathrm{e}{-6}$. Refer to Appendix E for an analysis of the impact of solver precision on the error in the mapping function used to estimate the gradient via the implicit function theorem. To obtain a good approximation of the times $t_-$ and $t_+$ to reach the boundary of the domain $\Omega$, we perform a binary search, between the time triggered by the stopping condition and the previous time before the event was triggered. There is no need for this binary search procedure to be differentiable thanks to the implicit differentiation theorem. To solve the eigenvalue problem, the values of $q$, $p$ and $w$ are computed along the field line $\gamma$ to obtain a spline which is dependent only on $t$ (avoiding numerous calls to the neural networks during the shooting phase). In these experiments, a piecewise linear function of 2000 parts is used to approximate these functions along the field line. The binary search is done with a tolerance of $1\mathrm{e}{-4}$ for the tabular experiments and $1\mathrm{e}{-8}$ for the image dataset. For all experiments, the number of eigenfunctions was fixed to 10. More details on architectures of the neural networks and the optimizer's parameters are provided in the appendix E.

### 5.2.1. EXPERIMENTAL RESULTS

**Local bases are tailored to each sample:** As a preliminary evaluation, to verify that DSL learns a different local basis for each example $x$, the local basis along the field line $\gamma^x(t)$ is analyzed for several different samples of the Dry Bean Dataset. As observed in figure 3, the local basis is different for the each of the examples represented, illustrating the local expressiveness of Deep Sturm–Liouville. As expected by the Sturm–Liouville theory, the boundaries satisfy the Dirichlet conditions and the $i^{th}$ base function crosses the $x$-axis exactly $i - 1$ times.

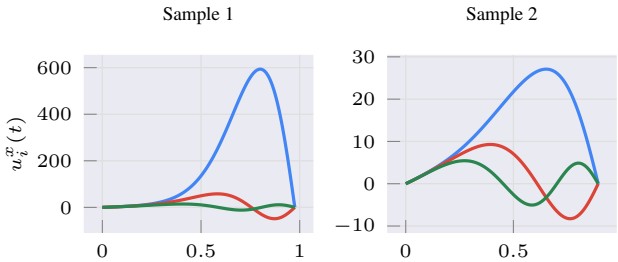

*Figure 3.* **Eigenfunctions on Dry Bean dataset.** For two samples, the first three eigenfunctions. The $x$-axis represents the time $t$ of the field line $\gamma^x(t)$.

**Impact of explicit and implicit regularization:** First, we perform a sensitivity analysis on the Dry Bean dataset (Figure 4) to evaluate the impact of implicit and explicit regularization on the classifier's performance. The results show that accuracy reaches its peak when the number of eigenvalues—serving as a parameter for implicit regularization—is around 10, achieving performance comparable to a MLP. As shown in the graph, DSL demonstrates robustness to this parameter, with accuracy remaining stable across a wide range of values. Significant accuracy drops are observed only when the spectral parameter is set too low or when the number of eigenvalues becomes excessively low.

**DSL achieves comparable performance than standard Neural Networks:** To demonstrate that Deep Sturm–Liouville can reach comparable performance to a Neural Network, DSL and NN were trained on several classification tasks. For this experiment, Neural Networks have similar architectures. Table 1 demonstrates that DSL achieves comparable results to NN with only 10 eigenfunctions.

**DSL achieves better sample efficiency:** To evaluate the impact of $1D$ regularization on generalization, we conducted a sample efficiency analysis to determine how effectively DSL can extract generalizable predictions from limited training data (see Fig.5). In this experiment, we trained both DSL and a standard Neural Networks (NN)

(a)
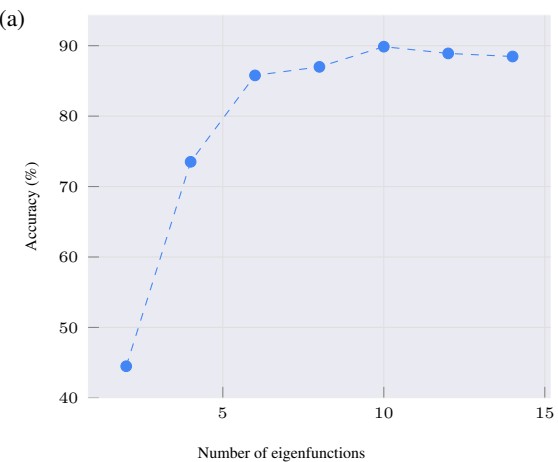

(b)
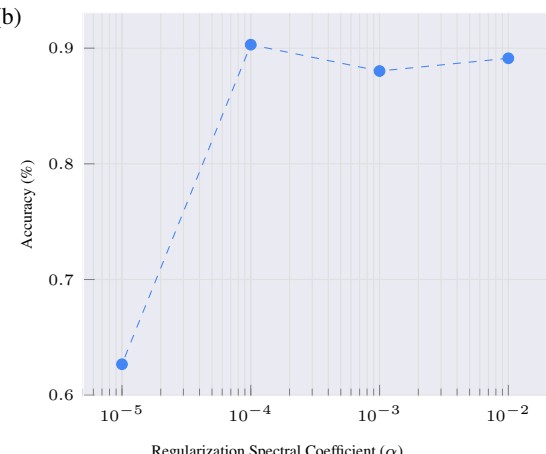

*Figure 4.* **Impact of the implicit and explicit regularization on Dry Bean dataset.** Validation accuracy as function of eigenfunctions basis size for the implicit regularization **(a)** and the spectral regularization coefficient $\alpha$ for explicit regularization in equation 9 **(b)**.

| DATA SET | DSL (OURS) | NODE |
|---|---|---|
| ADULT | 84.28% | 84.06% |
| DRY BEAN | 91.14% | 91.45% |
| BANK MARKETING | 83.10% | 83.77% |
| MNIST | 97.93% | 96.8%* |
| CIFAR10 | 58.38% | 58.9%* |

*Table 1.* **Evaluation.** Classification accuracies for DSL (Deep Sturm–Liouville) and NODE (Neural ODEs). * The results on Mnist and Cifar10 are sourced from Massaroli et al. (2020).

on small training sets, with sample sizes ranging from 100 to 800, and compared their test accuracy. Furthermore, as an ablation study, we investigate a modified DSL model in which the vector field $a(x)$ is not utilized. On the Bank dataset, as illustrated in Fig.5, DSL consistently outperformed traditional neural networks, showcasing its superior ability to generalize from fewer samples. Additional details about the experimental setup are provided in Appendix E.

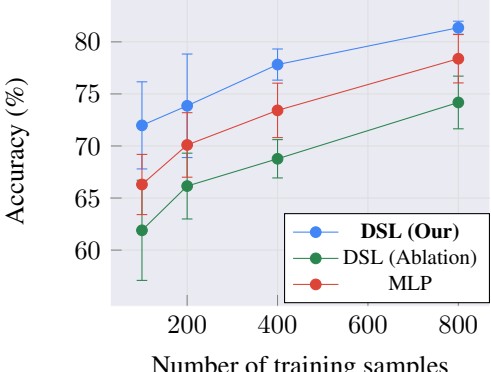

*Figure 5.* **Impact of training sample on Accuracy.** Test Accuracy on Bank dataset as function of number of training samples.

## 6. Limitations

Despite the promising results, Deep Sturm–Liouville suffers from several flaws. **Scalability.** Even if Deep Sturm–Liouville is scalable to high dimension, the gradient computation can be expensive due to the form of the problem to solve . It is not particularly due to the implicit differentiation theorem because we observe that the computation of the gradient on the weights of neural networks are the same order of magnitude than the computation of the jacobian on the eigenvalues and times to boundaries. Prediction computation can also be expensive. Even if the binary search itself is quick, the approximation of $p$, $q$, $w$ along the field line $\gamma$ can be costly despite their smoothness[2]. **Stability.** The estimation of the gradient of the ODE could be noisy if the solver is not precise enough. During training, in rare configurations, the ODE solver can fail to detect the boundary and the training fails.

## 7. Conclusion

A mathematical formulation has been developed to introduce the Sturm–Liouville Theory in the deep learning framework. We demonstrate the link between the Deep Sturm-Liouville formula and the Rank-1 Parabolic Eigenvalues problem. A trainable procedure based on implicit differentiation was implemented, successfully achieving comparable results to those of neural networks on tabular datasets, `MNIST` and Cifar10. We hope that our work paves the way for novel avenues in function regularization. Future work shall be done to develop the change of variable formula of DSL to obtain a generative classifier.

---

[2]Smoother functions allow for bigger step-size in numerical integration.

## Code Availability

The source code used to implement the models and conduct the experiments presented in this paper is publicly available at https://github.com/deel-ai-papers/deep-sturm-liouville.

## Acknowledgments

This work was carried out within the DEEL project,[3] which is part of IRT Saint Exupéry and the ANITI AI cluster. The authors acknowledge the financial support from DEEL's Industrial and Academic Members and the France 2030 program – Grant agreements n°ANR-10-AIRT-01 and n°ANR-23-IACL-0002.

## Impact Statement

This paper presents work whose goal is to advance the field of Machine Learning. There are many potential societal consequences of our work, none of which we feel must be specifically highlighted here.

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

## A. Eigenvalues Computation

First, we compute the lower and upper bounds of the eigenvalues of Sturm–Liouville Problem $[\lambda_n^-, \lambda_n^+]$ thanks to (Breuer & Gottlieb, 1971):

$$\lambda_n^+ \overset{\text{def}}{=} \frac{n^2\pi^2}{\min_t w(t)p(t) \cdot \left(\int_b^a \frac{1}{p(t)}dt\right)^2} + \max_t \frac{-q(t)}{w(t)}.$$

$$\lambda_n^- \overset{\text{def}}{=} \frac{n^2\pi^2}{\max_t w(t)p(t) \cdot \left(\int_b^a \frac{1}{p(t)}dt\right)^2} + \min_t \frac{-q(t)}{w(t)}.$$

(10)

Then we perform a binary search method to avoid tuning some hyper-parameters such as the learning rate if we had chosen the gradient descent.

Unfortunately, it is not possible to perform the binary search directly in the interval $[\lambda_i^-, \lambda_i^+]$. Indeed, the different intervals for each $\lambda_i$ may overlap meaning that there might be multiple eigenvalues $\lambda_j$ in the interval $[\lambda_i^-, \lambda_i^+]$. So the binary search is not guaranteed to find the correct eigenvalue $\lambda_i$.

To resolve this issue and to guarantee the monotonicity of the eigenvalue problem in the interval $[\lambda_i^-, \lambda_i^+]$, the Prüfer Substitution (Prüfer, 1926; Lebovitz, 2019) is used to ensure that there is a unique solution for each eigenvalue. The equations (3) are substituted by the following equations thanks to the change of variables:

$$\begin{cases} u_i(t) = r(t)\sin(\theta(t)), \\ \dfrac{du_i(t)}{dt} = \dfrac{r(t)}{p(t)}\cos(\theta(t)). \end{cases}$$

If $\lambda_n$ is the $n^{\text{th}}$ eigenvalue given by the Sturm–Liouville theorem, the equations can be re-expressed as:

$$\begin{aligned} \frac{d\theta(t)}{dt} &= (\lambda_n w(t) + q(t))\sin^2(\theta(t)) + \cos^2(\theta(t))\frac{1}{p(t)}, \\ \frac{dr(t)}{dt} &= \left[\frac{1}{p(t)} - (\lambda_n w(t) + q(t))\right]\frac{r(t)}{2}\sin(2\theta(t)), \\ \theta(a) &= 0, \quad \theta(b) = n\pi. \end{aligned}$$

(11)

The boundary conditions are dependent on the parameter $n$, relating to the $n^{th}$ eigenvalue, which is not the case with the initial formulation (3). This is what allows us to overcome the overlapping intervals problem.

Let $g(\lambda)$ be the function that maps each $\lambda$ to the value $\theta(b) - n\pi$ obtained by solving the equation (11) with the initial boundary condition $\theta(a) = 0$, for a given $\lambda$. The function $g$ is a strictly increasing function, so that a binary search can be applied between $[\lambda_i^-, \lambda_i^+]$ to find the $\lambda$ such that $g(\lambda) = 0$.

The computation of $\lambda_i^x$ is done in a similar way by using the shooting method along the field line $\gamma^x$ with binary search and by applying the Prüner substitution on equation (7).

## B. Gradient computation over $t_-^x, t_+^x$ and $\lambda_i$

To compute the gradient over the times to the boundaries and the eigenvalues, the implicit differentiation theorem is used. We define the mapping $H^{\theta,\lambda,t_-^x,t_+^x} : \Omega \to \mathbb{R}^{d+2}$, capturing the optimal conditions of the problem:

$$H_k^{\theta,\lambda,t_-^x,t_+^x}(x) = \begin{cases} u_k^{\theta,\lambda}(\gamma^x(t_+^x)), & \text{if } 1 \le k \le d, \\ \min_j \gamma_j^x(t_-^x), & \text{if } k = d+1, \\ \max_j \gamma_j^x(t_+^x) - 1 & \text{if } k = d+2. \end{cases}$$

*Remark* B.1. The two last conditions materialize the intersection of the field line $\gamma^x$ with $\partial\Omega$ for the specific domain $\Omega = ]0,1[^n$ that we use in our experiment. It should be defined differently for other convex domains such as the sphere.

*Remark* B.2. $\forall x \in \Omega, \quad H^{\theta,\lambda,t_-^x,t_+^x}(x) = 0.$

Following the implicit differentiation theorem (Krantz & Parks, 2012):

$$\nabla u_i^\theta(x) = \nabla_\theta u_i^\theta(x)$$
$$- \nabla_{\lambda,t_-,t_+} u_i^\theta(x) \mathbb{J}_{\lambda,t_-,t_+}^{-1} H^{\theta,\lambda,t_-^x,t_+^x}(x) \mathbb{J}_\theta H^{\theta,\lambda,t_-^x,t_+^x}(x).$$

## C. Proof Theorem 5.1

$u_i(x)$ form an orthogonal basis function on a open $\Omega$:

$$\int_\Omega v(x) u_i(x) u_j(x) dx = 0.$$

*Proof.* By Sturm–Liouville Theory, we have for all $x \in \Omega$ and for all $i \neq j$ :

$$\int_{t_-^x}^{t_+^x} w(\gamma^x(t)) u_i^x(t) u_j^x(t) dt = 0.$$

We will reformulate the integral over the field line $\gamma^x$ and by applying the line integrals change of variable and by (8):

$$\int_{\gamma^x} \frac{w(z)}{\|a(z)\|} u_i(z) u_j(z) d\mathcal{H}_\Omega^1(z) = 0.$$

We define the manifold $\partial\Omega_- \subset \mathbb{R}^n$ which is $(n-1)$-rectifiable:

$$\partial\Omega_- = \{\gamma^x(t_-^x) \quad \forall x \in \Omega\}.$$

By integrating over $\Omega_-$ we get:

$$\int_{\partial\Omega_-} \int_{\gamma^v} \frac{w(z)}{\|a(z)\|} u_i(z) u_j(z) d\mathcal{H}_\Omega^1(z) d\mathcal{H}_{\partial\Omega_-}^{n-1}(v) = 0.$$

We define:

$$P : \Omega \to \partial\Omega_-$$
$$P(x) = \gamma^x(t_-^x).$$

Since $P$ is Lipschitz we apply the co-area formula (Nicolaescu, 2011):

$$\int_\Omega \frac{w(x)}{\|a(x)\|} u_i(x) u_j(x) det|\mathbb{J}_P(x)| dx = 0.$$

We let:

$$v(x) = \frac{w(x)}{\|a(x)\|} det|\mathbb{J}_P(x)|.$$

Then:

$$\int_\Omega v(x) u_i(x) u_j(x) dx = 0.$$

$u_i(x)$ form an orthogonal basis functions under the weight function $v(x)$ on the domain $\Omega$. $\qquad \square$

## D. Proof Theorem 5.3

Uniform Deep Sturm–Liouville can be rewritten to a Dirichlet Rank-1 Parabolic Eigenvalues Problem when assuming that a(x) is the gradient of function with $a(x)_i > 0$.

*Proof.* From the equation (5), we will take the special case where:

$$A(x) = a(x)a^t(x).$$

Then we will develop the first component of the equation:

$$\nabla \cdot (a(x)a^t(x) \cdot \nabla u_i(x)) + q(x)u_i(x) = -\lambda_i w(x)u_i(x). \tag{12}$$

We will introduce the following change of variable $(t, \boldsymbol{y})$ such that:

$$v_i(t, \boldsymbol{y}) = u_i(x)$$
$$\nabla_x t = a(x)$$
$$\nabla_x y_k = a_{n_k}(x) \quad \forall k \in [1, n-1]$$

First, we show that such a change of variable exists:

Let $E$ be a one-dimensional real vector bundle over a manifold $M$. It is known (J. W. Milnor, 1974) that $E$ is trivial if and only if it admits a global nowhere-vanishing section—equivalently if its first Stiefel–Whitney class $w_1(E)$ vanishes. In this case, a nowhere-vanishing section $a$ provides a canonical trivialization $E \cong M \times \mathbb{R}$. Furthermore, if we choose local coordinates adapted to this trivialization and if there exists a smooth function $V : \mathbb{R}^n \to \mathbb{R}$ such that $\nabla V(x) = a(x)$, the Jacobian of the corresponding local map $T_x$ can be arranged so that its first row is precisely $a(x)$. This reflects the fact that the coordinate system is chosen to align the fiber direction (generated by $a(x)$) with the first coordinate axis. With an additional Riemannian metric on $M$, the orthogonal complement Bundles lemma (Lee, 2000) then provides an orthogonal decomposition $T_x M = \text{span}\{a(x)\} \oplus \text{span}\{a(x)\}^{\perp}$, completing the geometric picture. Consequently, the first row of the Jacobian is exactly $a(x)$, aligning the coordinate system with the direction defined by $a(x)$. Combined with the orthonormality of the remaining coordinates, this ensures that the first row of the Jacobian matrix is orthogonal to all other rows.

Then, we can expand $\nabla_x \cdot (a(x)a^t(x)\nabla_x u_i(x))$ as

$$
\begin{aligned}
=& \nabla_x \cdot \left( a(x)a^t(x) \left( a(x)\frac{\partial v_i(t, \boldsymbol{y})}{\partial t} + \sum_{k=1}^{n-1} \left( a_{n_k}(x)\frac{\partial v_i(t, \boldsymbol{y})}{\partial y_k} \right) \right) \right) \\
=& \nabla_x \cdot \left( \|a(x)\|^2 a(x)\frac{\partial v_i(t, \boldsymbol{y})}{\partial t} \right) \\
=& \|a(x)\|^2 a(x) \left( \frac{\partial^2 v_i(t, \boldsymbol{y})}{\partial t^2}a(x) + \sum_{k=1}^{n-1} \left( a_{n_k}(x)\frac{\partial^2 v_i(t, \boldsymbol{y})}{\partial y_k \partial t} \right) \right) \\
& + \|a(x)\|^2 \frac{\partial v_i(t, \boldsymbol{y})}{\partial t}\nabla_x \cdot a(x) \\
& + 2\left(\mathbb{J}_x a(x) \cdot a(x)\right) \cdot a(x)\frac{\partial v_i(t, \boldsymbol{y})}{\partial t} \\
=& \|a(x)\|^4 \frac{\partial^2 v_i(t, \boldsymbol{y})}{\partial t^2} \\
& + \left(2\left(\mathbb{J}_x a(x) \cdot a(x)\right) \cdot a(x) + \|a(x)\|^2 \nabla_x \cdot a(x)\right)\frac{\partial v_i(t, \boldsymbol{y})}{\partial t}
\end{aligned}
$$

We let:

$$b(x) = 2\left(\mathbb{J}_x a(x) \cdot a(x)\right) \cdot a(x) + \|a(x)\|^2 \nabla_x \cdot a(x)$$

Then because $a(x)_i > 0$, the equation (12) can be rewritten:

$$\Leftrightarrow \begin{cases} \|a(x)\|^4 \frac{\partial^2 v_i(t,\boldsymbol{y})}{\partial t^2} + b(x)\frac{\partial v_i(t,\boldsymbol{y})}{\partial t} + q(x)v_i(t,\boldsymbol{y}) = -\lambda_i w(x)v_i(t,\boldsymbol{y}) \\ \frac{dx}{dt} = a^{\circ-1}(x) \qquad \text{(Hadamard inverse of } a(x)) \end{cases} \tag{13}$$

$$\Leftrightarrow \begin{cases} \frac{\partial}{\partial t}\left(p(x)\frac{\partial v_i(t,\boldsymbol{y})}{\partial t}\right) + \tilde{q}(x)v_i(t,\boldsymbol{y}) = -\lambda_i \tilde{w}(x)v_i(t,\boldsymbol{y}) \\ \frac{dx}{dt} = \tilde{a}(x) \end{cases} \tag{14}$$

$\square$

# E. Experiments details

**Optimizer** The optimizers of `optax` library are adam (Kingma & Ba, 2014) with learning rate $2e{-}3$ for the tabular datasets and fromage (Bernstein et al., 2020) (lr=1e−2) for the MNIST and Cifar10 datasets. The losses are the hinge loss for the tabular datatsets and the categorical cross-entropy for the MNIST and Cifar10 datasets. The number of epochs is 40 for tabular dataset, 10 for Mnist and 80 for Cifar10.

**For the tabular datasets,** the functions $q(x)$, $\frac{1}{p(x)}$ and $w(x)$ are defined by a MLP with the features [128, 64, 32, 1] and leaky relu activations. The function $v(x) = 1$. The function $a(x)$ is a MLP [128, 64, 32, $k$] with tanh activations, where $k$ is the dimension of the input size of the data. All weights are initialized with glorot-uniform initializer.

**For the MNIST dataset**, the functions $q(x)$, $\frac{1}{p(x)}$ and $w(x)$ are defined by a convolutional neural network with [32,64,128] features and kernel (3,3), the features of the MLP are [32, 32, 16, 1] with tanh activations. The function $v(x) = 1$. The function $a(x)$ is an auto-encoder with [32,64] convolutions features and kernel (3,3) with 32 features and tanh activations. The weights are initialized with orthogonal initializer.

**For the CIFAR10 dataset**, the data are first projected by an encoder $e(x)$ in a latent space of dimension 128. The architecture of the encoder has [32, 64, 128] convolutions with kernel (3,3); the top of the encoder is a MLP [512, 256, 128]. The activation function is leaky-relu. To keep the projected domain within $[-0.5, 0.5]^{128}$ while avoiding vanishing gradients, the last activation of the encoder is defined as $h(x) = tanh(x/4.0)/2.0$. The encoder is learn on the side of other neural networks. The functions $q(x)$, $\frac{1}{p(x)}$ and $w(x)$ are defined by MLP with [128, 64, 66, 1] features and leaky-relu activations. The function $v(x)$ is a MLP with [128, 64, 66, 10] features and leaky-relu activation. The function $a(x)$ is a MLP with [256, 256, 256, 128] features and tanh activations. All weights are initialized with glorot-uniform initializer expect the MLP part of the encoder which are initialize with orthogonal initializer.

**Eigenvalues bounds** For the MNIST and Cifar10 datasets, to ensure that the eigenvalues are not too large at initialization, the output domain of each of the functions $a$, $p$, $q$ and $w$ was bounded. The choice of the appropriate bounds for each function is guided by the lower and upper bounds in the equations (10). To limit the eigenvalues to belong to the interval $\approx [-100, 100\ n^2\pi^2]$, the following constraints were implemented:

|        | $a(x)$      | $q(x)$      | $\frac{1}{p(x)}$ | $w(x)$      |
| ------ | ----------- | ----------- | ---------------- | ----------- |
| DOMAIN | $(0.01, 1)$ | $(-10, 10)$ | $(1, 10)$        | $(0.1, 10)$ |

*Remark* E.1. The eigenvalues bounds were taken experimentally to let enough range to the variation of the eigenvalues while maintaining a reasonable computation times.

These constraints are enforced by using sigmoid activations at the end of the model for $a$, $p$ and $w$ and a hyperbolic tangent activation for $q$.

**Eigenvalues regularization.** The value of the regularization coefficient of the equation (9) was fixed to $1e{-}4$.

**Domain and Data normalization.** As defined in the remark (B.1), the domain $\Omega$ is defined to be $]0, 1[^n$. Data are normalized so that they belong to $[0.25, 0.75]^n$, thus ensuring that they are included in $\Omega$, and that no example is too close to the boundary $\partial\Omega$, where the basis functions equal to 0 due to the Dirichlet conditions.

**Data Augmentation.** No data augmentation was performed.

**Baselines.** For MNIST and CIFAR-10, we report the results from Massaroli et al. (2020). For the tabular datasets, we used an architecture similar to the one used for the function $a(x)$.

**Low sample scenario.** We train DSL with strong spectral regularization ($\alpha = 10.0$) and perform experiments using 5 different random seeds. The best model is selected using a validation set of 1,000 samples, and testing is conducted on a separate test set of 1,000 samples. The model is trained for 200 epochs. "As part of an ablation study, to evaluate the role of the vector field in our approach, we apply the following Sturm–Liouville formulation:

$$F(x) = Lu(x, 0.5)$$

$$\frac{d}{dt}p(x,t)\frac{du(x,t)}{dt} + q(x,t)u(x,t) = \lambda(x)w(x,t)u(x,t)$$

$$u(x,0) = 0$$

$$\frac{du(x,0)}{dt} = 1$$

$$u(x,1) = 0$$

Here, $p$, $q$, and $w$ are modeled as MLPs. The input to each MLP is formed by concatenating the data point $x$ with the time variable $t$.

The following results show the accuracy of the ablation model evaluated on the tabular dataset:

| DATA SET | DSL (OURS) | DSL (ABLATION) | NODE |
|---|---|---|---|
| ADULT | 84.28% | 84.21% | 84.06% |
| DRY BEAN | 91.14% | 88.61% | 91.45% |
| BANK MARKETING | 83.10% | 84.63% | 83.77% |

**Solver sensitivity** We conduct a small experiment (Figure E) to analyze the sensitivity of the solver's precision on the mapping function used to estimate the gradient via the implicit function theorem. After achieving a certain level of precision, we observe that the error in the mapping function becomes linearly proportional to the solver's precision..

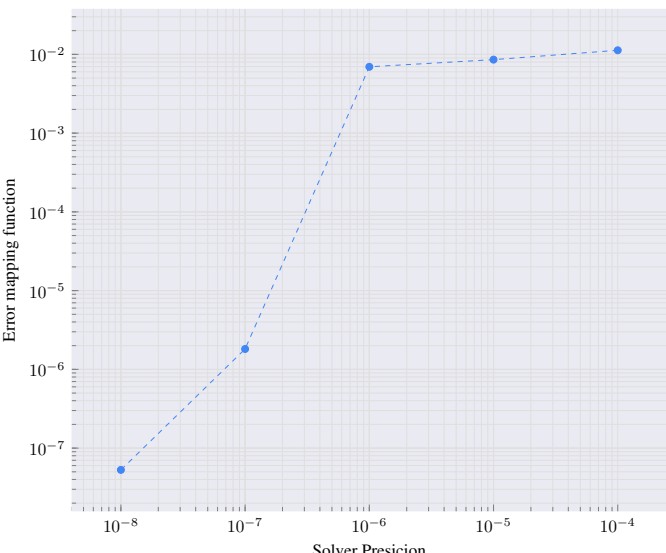

*Figure 6.* Error on the mapping function on the DryBean datatset

**Sample efficiency by eigenvalues:**

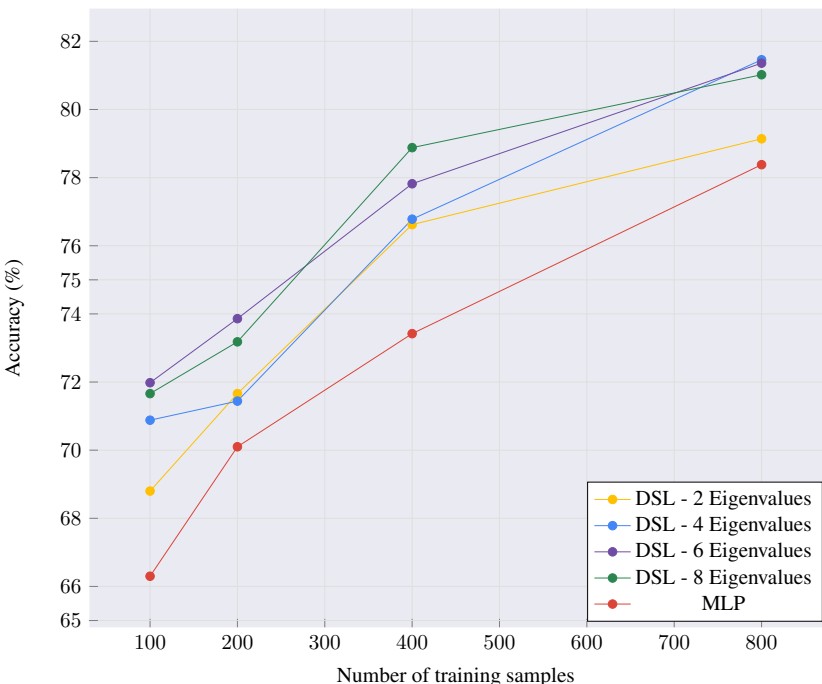

*Figure 7.* **Impact of training sample on Accuracy** Test Accuracy on Bank dataset as function of number of training samples and the number of eigenvalues.

