# OpenReview forum: "Deep Sturm–Liouville: From Sample-Based to 1D Regularization with Learnable Orthogonal Basis Functions"
_ICML.cc/2025/Conference — ICML 2025 poster_

### Official Review · Reviewer_cdFW · 2025-03-11

**Overall Recommendation:** 3

**Summary:**

This paper proposes a novel method (Deep Sturm-Liouville, DSL) that combines the Sturm-Liouville theorem with neural networks. By solving one-dimensional Sturm-Liouville problems, DSL generates orthogonal basis functions from the resulting eigenfunctions to approximate target functions and employs implicit gradients for network training. The method integrates both intrinsic implicit regularization and manually added explicit regularization. Experiments on the Adult, Dry Bean, Bank Marketing, MNIST, and CIFAR10 datasets demonstrate that DSL achieves performance comparable to traditional neural networks while exhibiting superior sample efficiency.

## update after rebuttal
I will maintain my score after reading the response from authors

**Claims And Evidence:**

In the abstract and introduction, the authors hypothesize that the generalization challenges of neural networks may stem from "0D regularization" (i.e., sample-point-based regularization) and claim that DSL can "overcome" this limitation. However:

**Theoretical gaps**: No rigorous proof is provided to show that DSL inherently achieves better regularization or generalization than conventional methods. The paper also fails to explain how 0D regularization negatively impacts neural network generalization.

**Experimental limitations**: The results only demonstrate comparable performance to existing methods (standard neural networks and neural ODEs), with no evidence that DSL actually addresses generalization difficulties.

**Essential References Not Discussed:**

No essential references were omitted.

**Experimental Designs Or Analyses:**

While the current results still meaningfully validate the feasibility and potential of DSL, the experimental section has certain limitations:

**Architectural ambiguity**: The neural network architectures (e.g., layer configurations, parameter counts) used for baseline comparisons are not clearly specified.

**Incomplete hyperparameter analysis**: While the impact of the number of basis functions is discussed, critical hyperparameters (e.g., the regularization coefficient α in Eq. 9, architectures of the field-line generation network) are not systematically analyzed.

**Computational fairness**: DSL requires additional time to compute basis functions and other intrinsic information. To ensure a fair comparison, the authors should include results for traditional neural networks trained under equivalent computational budgets.

**Methods And Evaluation Criteria:**

The methodological framework of DSL is aligned with the core objectives of this research.

**Other Comments Or Suggestions:**

Typos:

Page 2: "Liouville" is misspelled as "Liouiville."

Page 7, Line 333: "Dirichlet" is misspelled as "Dirichelt."

Figure references:

Figure 5 (Page 8) is not discussed in the main text. The analysis of sample efficiency in Section 5 should reference Figure 5 instead of Figure 7 in the appendix.

Page 7, Line 360: The reference to "Figure 5.2.1" is unclear.

**Other Strengths And Weaknesses:**

Have mentioned above.

**Questions For Authors:**

1. How exactly does the proposed method address the generalization challenges attributed to 0D regularization?

2. What is the mechanism behind the implicit regularization in DSL, and how does it theoretically or empirically outperform traditional neural network regularization?

**Relation To Broader Scientific Literature:**

Traditional regularization methods (e.g., weight decay, dropout) impose constraints in parameter space or via stochastic perturbations, while sample-level regularization (e.g., adversarial training, Mixup) enforces robustness to local input perturbations. These are inherently "0D" approaches, operating on discrete samples or parameters. In contrast, DSL introduces 1D regularization along field lines, extending constraints to continuous paths and implicitly enforcing function smoothness through orthogonal basis functions derived from Sturm-Liouville problems.

**Theoretical Claims:**

The theoretical proofs are rigorous.

---

> ### Author Rebuttal · Authors · 2025-03-31
>
> We would like to thank the reviewer for carefully reading our paper, providing thoughtful feedback and recognizing the novelty of our approach.
>
> I — In response to the reviewer's main concerns:
>
> 1 — 1D Regularization vs. 0D Regularization: By 0D regularization, we refer to regularization applied at discrete data points. For example, a term like $\sum_i \lvert \nabla_x F(x_i) \rvert$ computes the gradient of the predictor $F(x)$ only at the sampled points $x_i$. Ideally, one would compute a continuous regularization such as $\int_{\Omega} \lvert \nabla_x F(x) \rvert  dx$, but this is generally intractable. Our method bridges this gap by computing regularization along one-dimensional trajectories defined by a vector field $a(x)$ for each data point $x$. This 1D regularization provides a more informative approximation of the continuous case by integrating along these paths. Within this framework, we introduce two new types of regularization: spectral regularization and implicit regularization. These are complementary to traditional techniques (such as L1 or L2 regularization), which we do not aim to replace. In future work, our framework could also incorporate other forms of regularization or even modify the loss function itself.
>
> 2 — Implicit Regularization: The key idea behind implicit regularization in our framework is to retain only the first $n^\text{th}$ basis functions. By controlling the number of retained basis functions, we also control their level of oscillation—a direct consequence of the Sturm–Liouville theorem (see lines 173–180). This is analogous to the Fourier basis, where higher-order components exhibit more oscillation than lower-order ones. However, Sturm–Liouville provides a more general formulation. This property allows us to construct smoother function approximators, which have a known theoretical connection to generalization. Moreover, sample efficiency is inherently tied to generalization performance (see [1, 2] ). Our experiments on sample efficiency were specifically designed to assess generalization behavior. By limiting the number of basis functions, we directly control the expressiveness and smoothness of the learned function, thus influencing the regularization effect.
>
> II — Regarding the reviewer's concerns on "Experimental Design and Analyses":
>
> 1 — Architectural Ambiguity for the Baseline: The reviewer is right, and we will include a dedicated section to clarify this point in the appendix. For MNIST and CIFAR-10, we report the results from Massaroli et al. (2020), and we will explicitly reference this in the appendix. For the tabular datasets, we used an architecture similar to the one used for the function \( a(x) \). The source code is accessible via an anonymous GitHub https://rb.gy/n5iz02. Upon acceptance, the source code will be made publicly available to support the reproducibility of our work.
>
> 2 — Incomplete Hyperparameter Analysis: We conducted an ablation study in Figure 4(b) of the paper to analyze the impact of the spectral regularization coefficient on training accuracy. However, the coefficient $\alpha$ was missing from the figure and no reference was made to Equation (9). We will update the figure and its legend to include both the coefficient and the appropriate reference, as their absence is indeed misleading.
>
> 3 — Computational Fairness: The reviewer is right to highlight that computational budget is a critical factor for fair comparisons. For MNIST and CIFAR-10, we chose to report the results from Massaroli et al. (2020), a recognized reference. For the tabular datasets, we used similar architectures for both the baseline and the DSL model to ensure a fair comparison in terms of parameter count and model complexity.
>
> Lastly, thank you for pointing out several typos in the manuscript—we will correct them accordingly.
>
> [1] Zhang, Chiyuan, et al. "Understanding deep learning requires rethinking generalization." arXiv preprint arXiv:1611.03530 (2016).
>
> [2] Arpit, Devansh, et al. "A closer look at memorization in deep networks." International conference on machine learning. PMLR, 2017.

---

### Official Review · Reviewer_2moq · 2025-03-14

**Overall Recommendation:** 4

**Summary:**

In the present contribution authors describe a novel approximation scheme suitable for general mappings $\mathbb{R}^{m}\rightarrow \mathbb{R}^{n}$ where both $m$ and $n$ may be large. The scheme is suggested to be used as an alternative to neural networks.

A simplified description of the proposed mapping $y = f(x),$ where $x\in\mathbb{R}^{m},\,y\in\mathbb{R}^{n}$ is as follows
1. Solve the neural ordinary differential equation (ODE) starting from $x$ both for positive and for negative time, until the trajectory hits the boundary of the hypercube. From that record the whole trajectory $\gamma(t)$ and boundary points $t_{-}$, $t_{+}$, where the trajectory hit the hypercube.
2. Use $\gamma(t)$ to define parameters of one-dimensional Sturm–Liouville problem on the interval $\left[t_{-},t_{+}\right]$
3. Solve this problem for first $k$ eigenvectors $v_{j}(x),\,j=1,\dots,k$
4. Compute $y_{i} = \sum_{j}w_{ij} v_{j}(x),\,i=1,\dots,n$

In this scheme the following parameters are potentially learnable:
1. Vector field of neural ODE
2. Parameters of functions that compute coefficients of Sturm-Liouville problem from $\gamma(t)$
3. Number of basis functions used from Sturm-Liouville problem
4. Weights of linear transformation $w_{ij}$

Authors experimentally evaluated the proposed scheme and show that it achieves results competitive with other more standard approaches.

In addition to experimental evaluation authors prove two theoretical results:
1. The obtained eigenvectors are orthogonal on $\Omega\subset \mathbb{R}^{m}$, i.e., the input space where $x$ is defined and not only on the curve $\gamma(t)$.
2. That proposed scheme is related to the Dirichlet rank-1 parabolic eigenvalue problem.

## update after rebuttal

Summarised in https://openreview.net/forum?id=CzSNEvCckO&noteId=6lXg5LQbdT

**Claims And Evidence:**

Authors made several mild claims:
1. That the proposed approximation scheme leads to comparable performance to classical approaches.
2. Theoretical claims on orthogonality of eigenvalues.
3. Theoretical claims on the relation to parabolic eigenvalue problem.

I believe that experimental claims are supported by results on CIFAR and MNIST.

The first theoretical claim seems to be supported too. At least I can not point to any problems with the proof.

The second theoretical claim is not supported. I will provide more details in the appropriate section of the review.

**Essential References Not Discussed:**

I believe authors sufficiently discussed related literature. It seems Neural ODE is the most relevant related technique. Authors discuss Neural ODEs and perform numerical experiments explicitly comparing their method with Neural ODEs.

**Experimental Designs Or Analyses:**

I find the design and analysis of the experiment reasonable, besides the fact that no data on memory and computation load is available. Besides that in my view authors did not perform basic ablation study that seems appropriate given the complexity of the method they propose. I describe the suggested ablation below.

The scheme proposed by authors is to extract trajectory from neural ODE and later use this trajectory to construct a basis from Sturm-Liouville problem. The most obvious ablation to this scheme is to completely remove the step with ODE.

This can be done with simple modifications to Sturm-Liouville problem$$-\frac{d}{d\tau}\left(p_{\theta}(x, \tau)\frac{d}{d\tau} u_i(\tau)\right) + q_{\theta}(x, \tau)u_i(\tau) = \lambda_i \omega(x, \tau) u_{i}(\tau),$$where coefficients of Sturm-Liouville problem are neural networks.

Since authors implemented an efficient differentiation through Sturm-Liouville solver, this kind of replacement is a simple modification of their code.

I suggest authors perform this simplifications and report how this will affect accuracy and data efficiency.

**Methods And Evaluation Criteria:**

In general, I find evaluation criteria to be appropriate. However, I believe it would be beneficial for the readers to have access to more information on training time, memory load and other metrics along this line. The method proposed by the authors is quite complicated with many non-standard components that require custom derivation of derivative rules, this makes it hard to estimate computational requirements for using this technique.

**Other Comments Or Suggestions:**

I have several minor questions that I list here:
1. In several places authors mention that their approach is a "1D" regularisation and other known approaches are from "0D" regularisation, e.g., Lines 57-60. This terminology is not clear to me. Typically, regularisation implies some additional constraints, e.g., that the $l_2$ norm of weights is small, or that activations are normalised to $1$. Why does the approach by authors present a form of regularisation? If one uses some method to map between parts of input space (e.g., https://arxiv.org/abs/2303.16852), forming a trajectory between points, is it a form of regularisation too?
2. Lines 85-88, right column. "Conversely, DSL’s vector field operates directly within the input space ($\Omega\rightarrow\Omega$), ensuring that field lines traverse the entire domain." Why does the field traverse the entire domain? Since all components authors used are learnable, the trajectories of neural ODE can be arbitrary, not necessarily "dense" in $\Omega$.
3. Lines 303-306, right column. In the formulation of Theorem 5.1, indices $i$ and $j$ should be distinct which is not explicitly specified.

**Other Strengths And Weaknesses:**

**Strengths:**
1. Authors rigorously explain their approach including many details how derivatives are computed for the non-standard components.
2. To the best of my knowledge the approach is highly original with no directly related techniques in a published literature

**Weaknesses:**
1. The scheme is very complicated and it is not clear which parts are necessary since ablation is not available
2. The only advantage seems to be data efficiency that can be likely achieved with standard methods coupled with additional regularisation
3. The code is not available
4. Neural networks are still used to parametrise neural ODE and Sturm-Liouville problem, and in combination with minor to no improvement it rises the questions about the significance of the proposed approach

**Questions For Authors:**

Here a briefly summarise my main concerns:
1. Proof of Theorem 5.3 is not correct (see above).
2. The code is not available.
3. No information about computation and memory requirements.
4. The scheme is too complicated, and no ablation is available.

So, my main suggestions is to perform ablation study (see suggestions above) and correct the proof of Theorem 5.3.

**Relation To Broader Scientific Literature:**

If one considers MLP, each layer can be regarded as a progressive building of basis. The role of the last layer is to use this learned basis to solve the problem of interest with linear or simple nonlinear (link) function (see, e.g., Bishop CM, Nasrabadi NM. Pattern recognition and machine learning).

Present contributions suggest an alternative way to construct this data-dependent basis based on the initial-value problem and boundary-value problem.

The first related approach from the literature in neural ordinary differential equations https://arxiv.org/abs/1806.07366. Neural ODE are most widely used in generative modelling https://arxiv.org/abs/1810.01367, https://arxiv.org/abs/2210.02747. However, special classes of these models can also be applied directly to classification https://openreview.net/forum?id=SAv3nhzNWhw, anomaly detection https://arxiv.org/abs/2302.07253, reduced-order modelling https://www.nature.com/articles/s41598-023-36799-6, segmentation https://arxiv.org/abs/2502.06034, etc.

Since the technique proposed in the present contribution is related to partial-differential equations. This provides a second link to existing literature.

Methods based on PDEs were popular in the image processing problems, e.g., Chan TF, Shen J, Vese L. Variational PDE models in image processing, but they fall out of favour with the adoption of methods based on neural networks. PDE-based approaches also recently made they way into the deep learning, e.g., in https://arxiv.org/abs/2403.15726 authors used reaction-diffusion model, in https://arxiv.org/abs/2502.06034 a wave equation, and in http://proceedings.mlr.press/v107/sun20a/sun20a.pdf a general neural PDE is considered. Still, PDE-based approaches are rare and the method proposed by authors of the present contribution is an interesting exception.

**Theoretical Claims:**

I reviewed proofs for both theoretical claims.

The first one of the orthogonality of the eigenvalues "in the bulk" seems fine (Appendix C).

The proof for the second theoretical statement is not correct. In Appendix D on lines 728-732 authors assume the coordinate transformation they define is valid. Unfortunately, the assumption of the theorem does not exclude the situation when this transformation is not possible to define.

In the theorem authors assume $a_i(x)>0$. Next, they define scalar coordinate $t(x)$ by identity $\nabla_{x} t(x) = a(x)$. To be able to do that, one need $a(x)$ to be conservative field. If this is not assumed one can easily build pathological examples. Consider$$a(x) = \frac{1}{2}\begin{pmatrix}(x_1 - x_2)^2\\\\(x_1 + x_2)^2\\\\1\end{pmatrix}.$$Curl of this field reads$$\nabla_{x}\times a(x) = \begin{pmatrix}0\\\\0\\\\ 2 x_1\end{pmatrix}.$$Since $a_i(x) >0$ holds our choice agrees with conditions states in the theorem. However, it is not possible that $a(x) = \nabla_x t(x)$ for some scalar function $t(x)$ since curl of $a(x)$ is not zero.

The rest of the coordinates defined by the author also have this property but for vectors $a_{k}(x)$ that are orthogonal to $a(x)$. This means these fields should also be potential, and authors need to carefully explain how they are going to build these additional orthogonal fields. The reference to the Gram-Schmidt process is not sufficient.

---

> ### Author Rebuttal · Authors · 2025-03-31
>
> The authors would like to thank you for your detailed review, for carefully examining the demonstrations and for recognizing the novelty of our approach.
>
> I — Demonstration 2
>
> First, we would like to address the mistake you identified in Demonstration 2, specifically in lines 728–732. The reviewer is absolutely correct, and we sincerely thank you for pointing this out. This demonstration is a key element of our paper, and we greatly appreciate your attention to its details. Fortunately, the mistake can be corrected without affecting the main results. Below, we explain how to fix it. As a reminder, the objective of this part of the demonstration is to construct a change of variables such that the first row of the Jacobian matrix of the mapping function $(t, y = F(x))$ is the vector $a(x)$ which is required to be orthogonal to the remaining rows of the Jacobian.
>
> (i) — As the reviewer correctly pointed out, $a(x)$ must be the gradient of a scalar function. We will explicitly add this as a condition in the theorem. This is directly related to lines 251–252 of the paper, where we assume the absence of singular points.
>
> (ii) — The reviewer also rightly pointed out an issue in the construction of the orthogonal basis: the Gram-Schmidt process alone is not enough as the resulting vectors also need to be gradients of scalar functions. To address this, we will leverage tools from differential geometry by interpreting the change of variables as defining a one-dimensional bundle generated by the vector field $a(x)$. We will add the following explanation to the paper:
>
> ``Let \(E\) be a one-dimensional real vector bundle over a manifold \(M\). It is known (see Milnor \& Stasheff's Characteristic Classes) that \(E\) is trivial if and only if it admits a global nowhere‐vanishing section—equivalently if its first Stiefel–Whitney class \(w_1(E)\) vanishes. In this case, a nowhere‐vanishing section \(a\) provides a canonical trivialization $E \cong M \times \mathbb{R}$. Furthermore, if we choose local coordinates adapted to this trivialization and if there exists a smooth function $V : \mathbb{R}^n \rightarrow \mathbb{R}$ such that $\nabla V(x)=a(x)$, the Jacobian of the corresponding local map \(T_x\) can be arranged so that its first row is precisely \(a(x)\). This reflects the fact that the coordinate system is chosen to align the fiber direction (generated by \(a(x)\)) with the first coordinate axis. With an additional Riemannian metric on \(M\), the orthogonal complement Bundles lemma (see John M. Lee, Introduction to Smooth Manifolds) then provides an orthogonal decomposition $ T_xM = \operatorname{span}\{a(x)\} \oplus \operatorname{span}\{a(x)\}^\perp, $ completing the geometric picture. Consequently, the first row of the Jacobian is exactly $a(x)$, aligning the coordinate system with the direction defined by $a(x)$. Combined with the orthonormality of the remaining coordinates, this ensures that the first row of the Jacobian matrix is orthogonal to all other rows.''
>
> II — Ablation study
>
> To evaluate the role of the vector field in our approach, the reviewer suggested an ablation study that removes its influence. We agree this is a valuable experiment and would like to describe it in more detail. In this ablation, we apply the proposed Sturm–Liouville formulation over the time interval $[0, 1]$ and evaluate the basis functions at $t = 0.5$. The input to each MLP is formed by concatenating the data point $x$ with the time variable $t$.
>
> For the sample efficiency results and the accuracy performance on tabular data, please find below the updated plot and table that includes the ablation model https://rb.gy/xjx9tk. Our results show that DSL achieves higher sample efficiency and comparable accuracy compared to the ablation model.
>
> III — Weakness and other questions
>
> 1 — Code Availability: The source code is accessible via an anonymous GitHub https://rb.gy/n5iz02. Upon acceptance, the source code will be made publicly available.
>
> 2 — Use of Neural Networks in DSL: We acknowledge the dependency of the DSL on neural networks. As a direction for future work, we are exploring ways to design a version of the DSL that does not rely on neural networks. DSL represents a foundational block that we aim to develop further.
>
> 3 — 0D vs 1D Regularization: Since this question was also raised by another reviewer, we addressed it in the rebuttal of Reviewer cdFW (1).
>
> 4 — DSL Coverage of $\Omega$: The reviewer is right to point out that, for a general vector field, the trajectories might not cover the entire domain $\Omega$. To ensure full coverage, the ODE must satisfy certain constraints. These are discussed in the paper between line 215 (second column) and line 253 (first column), where we explain the uniqueness of $\gamma(t_-)$ and $\gamma(t_+)$. However, we realize this section may not clearly state that these constraints also guarantee coverage of $\Omega$. We will add an explicit sentence to highlight this point.

---

> > ### Comment · Reviewer_2moq · 2025-04-02
> >
> > I would like to thank the authors for a detailed reply and additional ablation study. In the rebuttal authors addressed my major concerns so I changed my recommendation accordingly.

---

### Official Review · Reviewer_upBa · 2025-03-14

**Overall Recommendation:** 3

**Summary:**

This paper introduces Deep Sturm-Liouville (DSL), a novel function approximator that integrates the Sturm-Liouville Theorem (SLT) into deep learning to achieve continuous 1D regularization along field lines in the input space. Demonstrates competitive performance and improved sample efficiency on diverse datasets including MNIST and CIFAR-10, showing DSL's effectiveness in practical machine learning tasks.

**Claims And Evidence:**

After careful review, I did not find any claims that were obviously problematic or lacked sufficient support. The evidence provided seems comprehensive and convincing. The theoretical analysis is rigorous, and the experimental verification validates the authors' theoretical analysis.

**Essential References Not Discussed:**

No literature missing.

**Experimental Designs Or Analyses:**

The experimental designs and analyses in the paper are generally sound and valid.

**Methods And Evaluation Criteria:**

The proposed methods and evaluation criteria are appropriate.

**Other Comments Or Suggestions:**

No

**Other Strengths And Weaknesses:**

Strengths:
The paper presents a novel integration of Sturm-Liouville theory with deep learning, creating the Deep Sturm-Liouville (DSL) framework. The introduction of 1D regularization along field lines in the input space addresses a fundamental limitation of traditional sample-based (0D) regularization methods, providing a new dimension for controlling model complexity and improving generalization. The experiment verifies the correctness of the proposed method. The overall structure of the paper is clear and easy for readers to understand.

Weaknesses:
1、	The computational overhead of solving Sturm-Liouville problems might restrict scalability to very large-scale problems or real-time applications, especially in scenarios with abundant training data.
2、	The experimental is limited. The author only verified it on small datasets. There is a lack of verification on the large-scale datasets.

**Questions For Authors:**

1、	The computational overhead of solving Sturm-Liouville problems might restrict scalability to very large-scale problems or real-time applications, especially in scenarios with abundant training data.
2、	The experimental is limited. The author only verified it on small datasets. There is a lack of verification on the large-scale datasets.

**Relation To Broader Scientific Literature:**

The paper's integration of Sturm-Liouville Theory with deep learning provides a novel mathematical foundation for developing more expressive and generalizable function approximators. This interdisciplinary approach bridges gaps between applied mathematics and machine learning, offering new insights into how to design models that can better capture the underlying structure of data.

**Theoretical Claims:**

I did not check all the details of the proof line by line, but on the whole, the theoretical analysis part of the paper has clear logic and reasonable proof structure.

---

> ### Author Rebuttal · Authors · 2025-03-31
>
> We thank the reviewer upBa for taking the time to review our paper in detail and for recognizing the novelty of our approach. The novelty of our approach was highlighted also by reviewer 2moq, who described our article as "highly original with no directly related techniques in a published literature". We appreciate your concern regarding the scalability of our method to large-scale datasets or real-time applications. We fully agree that this limitation may hinder the immediate widespread deployment of our framework in such contexts.
>
> However, as the reviewer noted, the core contribution of our work lies in introducing a new and principled approximator framework for regularization, grounded on theoretical results. As such, we believe it should be viewed as a foundational step—laying the groundwork for future improvements and extensions.
>
> Improving scalability is a clear direction for future work. Potential avenues include designing more efficient approximations of the Sturm–Liouville basis, new methods to compute gradients through the eigenvalue process that avoid relying on the implicit differentiation theorem, as well as exploring alternative ODE solvers that are better suited to Sturm–Liouville-type equations. Additionally, we plan to explore new forms of regularization beyond the spectral and implicit ones introduced here. We see this work as the first step in a broader research agenda that combines structure, theory, and learning in a unified framework. We will update the paper to make this limitation clearer—especially in the limitations section—and outline these directions as promising areas for future work.

---

### Decision · Program_Chairs · 2025-05-01

**Decision:**

Accept (poster)

**Comment:**

This paper introduces a novel class of function approximators called Deep Sturm-Liouville, which enables 1D regularization along continuous trajectories. The reviewers generally agree that this paper presents an interesting new approach to function approximation (even if at the cost of additional computational overhead), which is justified both theoretically and empirically. Based on this consensus, I recommend acceptance of the paper.